

# Genetic diversity of pomegranate germplasm collection from Spain determined by fruit, seed, leaf and flower characteristics

Juan J. Martinez-Nicolas[1], Pablo Melgarejo[1], Pilar Legua[1], Francisco Garcia-Sanchez[2] and Francisca Hernández[1]

[1] Producción Vegetal y Microbiología, Universidad Miguel Hernández de Elche, Orihuela, Spain
[2] Department of Plant Nutrition, CEBAS-CSIC, Spain

Corresponding author
Juan J. Martinez-Nicolas,
juanjose.martinez@umh.es

## ABSTRACT

**Background.** Miguel Hernandez University (Spain) created a germplasm bank of the varieties of pomegranate from different Southeastern Spain localities in order to preserve the crop's wide genetic diversity. Once this collection was established, the next step was to characterize the phenotype of these varieties to determine the phenotypic variability that existed among all the different pomegranate genotypes, and to understand the degree of polymorphism of the morphometric characteristics among varieties.

**Methods.** Fifty-three pomegranate (*Punica granatum* L.) accessions were studied in order to determine their degree of polymorphism and to detect similarities in their genotypes. Thirty-one morphometric characteristics were measured in fruits, arils, seeds, leaves and flowers, as well as juice characteristics including content, pH, titratable acidity, total soluble solids and maturity index. ANOVA, principal component analysis, and cluster analysis showed that there was a considerable phenotypic diversity (and presumably genetic).

**Results.** The cluster analysis produced a dendrogram with four main clusters. The dissimilarity level ranged from 1 to 25, indicating that there were varieties that were either very similar or very different from each other, with varieties from the same geographical areas being more closely related. Within each varietal group, different degrees of similarity were found, although there were no accessions that were identical. These results highlight the crop's great genetic diversity, which can be explained not only by their different geographical origins, but also to the fact that these are native plants that have not come from genetic improvement programs. The geographic origin could be, in the cases where no exchanges of plant material took place, a key criterion for cultivar clustering.

**Conclusions.** As a result of the present study, we can conclude that among all the parameters analyzed, those related to fruit and seed size as well as the juice's acidity and pH had the highest power of discrimination, and were, therefore, the most useful for genetic characterization of this pomegranate germplasm banks. This is opposed to leaf and flower characteristics, which had a low power of discrimination. This germplasm bank, more specifically, was characterized by its considerable phenotypic (and presumably genetic) diversity among pomegranate accessions, with a greater proximity existing among the varieties from the same geographical area, suggesting that over time, there had not been an exchange of plant material among the different cultivation areas. In summary, knowledge on the extent of the genetic diversity of the

collection is essential for germplasm management. In this study, these data may help in developing strategies for pomegranate germplasm management and may allow for more efficient use of this germplasm in future breeding programs for this species.

# INTRODUCTION

Pomegranate is a deciduous fruit tree, and its cultivation has been greatly expanded into several countries in recent years, especially those with a Mediterranean-like climate. In Spain, for example, the total acreage used today for its cultivation is about 2,791 ha, with an annual production of about 43,324 metric tons (*MAGRAMA, 2014*). The growing interest in this fruit is not only due to the fact that it is pleasant to eat, but it is also because it is considered to be a functional product that has been shown to be beneficial to human health, as it contains several types of substances that are useful in disease prevention (*Melgarejo & Artés, 2000*; *Melgarejo & Salazar, 2003*; *Cam, Hisil & Durmaz, 2009*; *Legua et al., 2012*; *Zaouay et al., 2012*; *Calani et al., 2013*; *Melgarejo-Sánchez et al., 2015*). Therefore, the demand for this fruit has increased in the last 10 years, as it has been used in industrial processing to obtain pomegranate juice, jams, vegetable extracts, etc. (*Melgarejo-Sánchez et al., 2015*).

The pomegranate's place of origin is considered to be Central Asia, from where it has spread to the rest of the world (Mediterranean Basin, Southern Asia and several countries of North and South America). It is a temperate-climate species that requires high temperatures to mature properly, but it is also easily spread in arid and semi-arid areas of the world, as it is tolerant to salinity and water scarcity, factors that usually limit the growth of other agronomical crops in these areas. Its successful adaptation to abiotic stress conditions, which characterize the Spanish Mediterranean climate, has led to its wide dispersion in this geographical area and to the appearance of a multitude of new, local individuals over time beginning with specific varieties (*Naeini et al., 2004*; *Naeini, Khoshgoftarmanesh & Fallahi, 2006*; *Martínez et al., 2006*; *Sarkhosh et al., 2006*).

These new varieties have been grouped under the same denomination, however, each one of them could have different agronomic characteristics as compared to their original progenitor. For example, *Melgarejo & Salazar* (*2003*) observed that under the denomination "Mollar de Elche" (ME) there were varieties with different agronomic characteristics. In order to better identify the fruit, *Verma, Mohanty & Lal* (*2010*) have mentioned the importance of agronomically-characterizing of varieties of a specific cultivar from the place where they originated to the areas where they disseminated, as being useful for understanding the evolution of the cultivar and for maintaining the biodiversity of the varieties, as well as for the improvement of agronomic characteristics of the crops.

In 1992, the Miguel Hernandez University created a germplasm bank of the varieties of pomegranate found in Southeastern Spain in order to preserve the crop's wide genetic diversity. Since its creation, many local types have been inventoried, described and planted in the experimental farm at EPSO (Escuela Politécnica Superior de Orihuela- Miguel

Hernandez University, Alicante). Currently, the collection contains 59 accessions that have been collected from different growing areas in Spain, representing about 16 local denominations (*Melgarejo*, *1993*). Once this collection was established, the next step was to determine its genetic biodiversity, and to classify the germplasm bank according to their agronomic characteristics rather than according to only a botanical point of view, as pomegranate consumption is important in both the fresh-fruit market and the processing industry. For this, the evaluation of the different morphometric and fruit characteristics was necessary, as this would a better describe and compare the genetic diversity of this germplasm collection. *Mars & Marrakchi* (*1999*) revealed the usefulness of measuring morphometric and chemical compound fruit variables such as weight, length, diameter, external color, seed number, length and diameter of the calyx, juice's volume, color, pH, total soluble solids TSS (g/l) and total acidity TA (g/l), in order to determine the genetic diversity of a pomegranate germplasm bank in Tunisia, composed of thirty pomegranate (*Punica granatum* L.) accessions.

To study the genetic diversity of the germplasm bank, microsatellite markers have also been used. *Singh et al.* (*2015*) validated the efficiency of this molecular tool on a pomegranate collection comprised of 88 accessions (37 domesticated and 51 wild). The study measured the structure of the population among the wild and domesticated accessions. *Ophir et al.* (*2014*), in a study using Single Nucleotide Polymorphism (SNP) Markers on 105 worldwide pomegranate accessions, located in the pomegranate germplasm collection at the Newe Ya'ar Research Center in northern Israel, observed that genetic diversity was primarily due to the geographic location of origin.

In the present work, we focus on morphometric and chemical compound measurements that will allow us to gain basic but needed knowledge on the agronomic characteristics of this pomegranate germplasm collection (grown under homogeneous conditions). If the phenotypic variability is found to be high, then the assumption is made that they are also genotypically different. These results could lead us to further characterize this collection through genetic analysis.

In our study, aside from the parameters mentioned above (*Mars & Marrakchi*, *1999*), parameters related to seeds, leaves and flowers were also measured, in order to have more complete information for determining the phenotypic and presumably genetic diversity among all the accessions. Therefore, the objective of this research was to determine the phenotypic variability that exists among all the different genotypes of Southeastern Spanish pomegranate, to understand the degree of polymorphism of the morphometric characteristics among varieties, and to establish the existing variability among accessions from the same family. Also, this research work had the advantage that the data used were taken on three consecutive years from trees that were planted in the same field thereby avoiding any edaphoclimatic effect on the results.

## MATERIAL AND METHODS

### Plant material

The areas prospected and the germplasm collecting procedures were as reported in *Melgarejo* (*1993*). Fifty-three accessions, representing 16 denominations, were included in

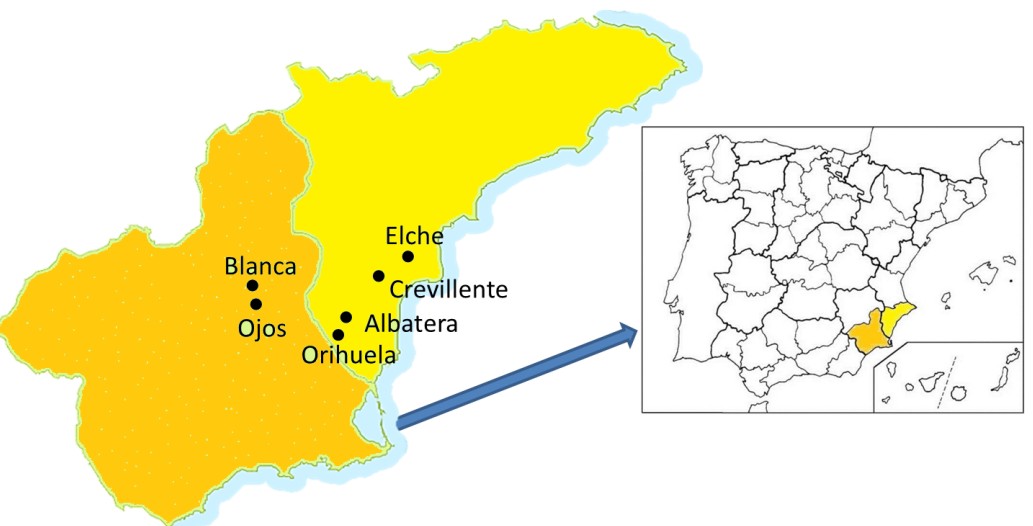

| Blanca | Albatera |
|---|---|
| AB1, SFB1, PB1 | BA1, MA1, MA2, MA3, MA4, MA5 |
| **Crevillente** | **Ojos:** |
| MC1, VA1 | ADO2, ADO3, BO1, CRO1, CRO2, PDO2, PTO2, PTO3, PTO4, PTO5, PTO6, PTO7, PTO8 |
| **Elche** | **Orihuela** |
| ME1, ME2, ME3, ME3.1, ME4, ME5, ME6, ME7, ME8, ME9, ME10, ME11, ME12, ME13, ME14, ME15, ME16, ME17, ME18, ME19, ME20, ME21 | MO2, MO3, MO4, MO5, MO6 |

**Figure 1 Location of the areas of origin of the accessions that comprised the germplasm collection studied.** The name of each accession according to the codes used can be found in Table 1.

the present study (Fig. 1 and Table 1). They were represented by twenty-five year old adult trees, maintained within the same collection in Alicante in the Southeast region of Spain (*Melgarejo*, *1993*).

Pomegranate trees were grown under homogeneous conditions in a loamy clay soil with a spacing of 5 × 4 m. A drip irrigation system was used for fertigation purposes. The collection was located in the experimental orchards belonging to the Miguel Hernández University, located in the province of Alicante, Spain (latitude: 38°03′50″N, longitude: 02°03′50″W and an altitude of 26 m above sea level). According to *Papadakis' (1966)* classification, the experimental plot had a subtropical Mediterranean climate. The annual mean temperature was 19 °C, with mild winters (11 °C in January) and hot summers (28 °C in August). A low annual precipitation of 300 mm was recorded, mostly falling in spring and autumn.

## Characters studied

The studies were based on measuring the characteristics of fruits, seeds, leaves and flowers. Morphometric measurements and chemical analyses were carried out on samples from 20 mature fruits, 25 seeds, 50 leaves and 25 flowers from each tree per variety, using a total of four trees per variety. The study was conducted over three consecutive years, and the following variables were measured:

**Table 1  Names, abbreviations and origin of pomegranate accessions evaluated.**

| Code | Accession | Location | Code | Accession | Location |
|------|-----------|----------|------|-----------|----------|
| AB1 | Albar de Blanca 1 | Blanca (Murcia) | ME13 | Mollar de Elche 13 | Elche (Alicante) |
| ADO2 | Agridulce de Ojós 2 | Ojos (Murcia) | ME14 | Mollar de Elche 14 | Elche (Alicante) |
| ADO3 | Agridulce de Ojós 3 | Ojos (Murcia) | ME16 | Mollar de Elche 16 | Elche (Alicante) |
| BA1 | Borde de Albatera 1 | Albatera (Alicante) | ME17 | Mollar de Elche 17 | Elche (Alicante) |
| BO1 | Borde de Ojós 1 | Ojos (Murcia) | ME18 | Mollar de Elche 18 | Elche (Alicante) |
| CRO1 | Casta del Reino 1 | Ojos (Murcia) | ME19 | Mollar de Elche 19 | Elche (Alicante) |
| CRO2 | Casta del Reino 2 | Ojos (Murcia) | ME20 | Mollar de Elche 20 | Elche (Alicante) |
| MA1 | Mollar de Albatera 1 | Albatera (Alicante) | ME21 | Mollar de Elche 21 | Elche (Alicante) |
| MA2 | Mollar de Albatera 2 | Albatera (Alicante) | MO2 | Mollar de Orihuela 2 | Orihuela (Alicante) |
| MA3 | Mollar de Albatera 3 | Albatera (Alicante) | MO3 | Mollar de Orihuela 3 | Orihuela (Alicante) |
| MA4 | Mollar de Albatera 4 | Albatera (Alicante) | MO4 | Mollar de Orihuela 4 | Orihuela (Alicante) |
| MA5 | Mollar de Albatera 5 | Albatera (Alicante) | MO5 | Mollar de Orihuela 5 | Orihuela (Alicante) |
| MC1 | Molar de Crevillente | Crevillente (Alicante) | MO6 | Mollar de Orihuela 6 | Orihuela (Alicante) |
| ME1 | Mollar de Elche 1 | Elche (Alicante) | PB1 | Piñonenca de Blanca 1 | Blanca (Murcia) |
| ME2 | Mollar de Elche 2 | Elche (Alicante) | PDO2 | Piñón duro de Ojós 2 | Ojos (Murcia) |
| ME3 | Mollar de Elche 3 | Elche (Alicante) | PG | Puente Genil | Puente Genil (Córdoba) |
| ME3.1 | Mollar de Elche 3.1 | Elche (Alicante) | PTB1 | Piñón tierno de Blanca 1 | Blanca (Murcia) |
| ME4 | Mollar de Elche 4 | Elche (Alicante) | PTO2 | Piñón tierno de Ojós 2 | Ojos (Murcia) |
| ME5 | Mollar de Elche 5 | Elche (Alicante) | PTO3 | Piñón tierno de Ojós 3 | Ojos (Murcia) |
| ME6 | Mollar de Elche 6 | Elche (Alicante) | PTO4 | Piñón tierno de Ojós 4 | Ojos (Murcia) |
| ME7 | Mollar de Elche 7 | Elche (Alicante) | PTO5 | Piñón tierno de Ojós 5 | Ojos (Murcia) |
| ME8 | Mollar de Elche 8 | Elche (Alicante) | PTO6 | Piñón tierno de Ojós 6 | Ojos (Murcia) |
| ME9 | Mollar de Elche 9 | Elche (Alicante) | PTO7 | Piñón tierno de Ojós 7 | Ojos (Murcia) |
| ME10 | Mollar de Elche 10 | Elche (Alicante) | PTO8 | Piñón tierno de Ojós 8 | Ojos (Murcia) |
| ME11 | Mollar de Elche 11 | Elche (Alicante) | SFB1 | San Felipe de Blanca 1 | Blanca (Murcia) |
| ME12 | Mollar de Elche 12 | Elche (Alicante) | VA1 | Valenciana de Albatera 1 | Albatera (Alicante) |

### Fruits

Fruit weight (FW), expressed in g; equatorial diameter (FD1), expressed in mm; calyx diameter (FD2), expressed in mm; fruit height without calyx (FL1), expressed in mm; total fruit height (FL2), expressed in mm; calyx height (FL3), expressed in mm; number of carpels (Nc) counted on the equatorial section; rind weight plus weight of carpellary membranes (PcMc), expressed in g; skin thickness (Ec), expressed in mm (measurements were performed on two opposite sides in the equatorial plane); aril yield calculated as (Rs) = [FW − (PcMc)/FW] × 100 (%).

Diameters, fruit height and skin thickness were measured with an electronic digital slide gauge (Mitutoyo), accurate to 0.01 mm. Fruit weights and Rind weight plus weight of carpellary membranes were measured with a digital scale (Sartorius Model BL-600) accurate to 0.1 g.

### Arils

After extracting the seeds by hand, 25 of them were randomly chosen from a homogenized sample in every sampling year. The following seed characteristics were studied (*Martínez*

*et al.*, *2006*): maximum width (Sw) and length (SL), measured with a digital caliper (Mitutoyo) accurate to 0.01 mm; aril weight (SW), determined with a precision weighing device (Mettler AJ50) accurate to 0.0001 g; juice volume (JV), using an electric extractor and a seed sample of 100 g; total soluble solids (TSS) (°Brix), determined with an Atago N-20 refractometer at 20 °C; total acidity, expressed as citric acid (AT), determined with an acid–base potentiometer and pH; and maturity index (MI = TSS/TA).

The most current classification that has been established for Spanish varieties (*Melgarejo*, *1993*) were used: Sweet varieties: MI = 31–98; Sour-sweet varieties: MI = 17–24; Sour varieties: MI = 5–7. Three repetitions per clone and year were carried out.

### Seeds

The parameters measured in the seeds (woody portion) were: maximum width ($w$) and length ($l$), measured with the same digital caliper as above; weight of the woody portion (wpw) of each seed using the above-mentioned precision balance; woody portion index (wpi), determined from the wpw/SW ratio 100 (%).

### Leaves

The leaves studied were collected in September, by choosing 50 adult leaves per tree, normal and leaves that sprouted in the spring. This sampling was done in the four cardinal directions of the tree. The length measurements of the leaves were performed with a digital caliper (Mitutoyo) accurate to 0.01 mm. The leaf surface area was determined with an image analyzer "Digital Image Analysis System" Delta-T model. The measured variables were: LW, leaf width (mm); Ll, blade length (mm); Lt, total length of the leaf (mm); Lp, petiole length (mm); LS, leaf surface area ($mm^2$).

### Flower

The flowers were randomly sampled during the period of full flowering in the mid-May, taking a total of 25 flowers per tree from four trees. This sampling was done in the four cardinal directions of the tree. Length measurements were performed using a digital caliper (Mitutoyo) accurate to 0.01 mm. The measured variables were: FD, flower diameter (mm); FL, flower length (mm); NP, number of petals; Ns, number of sepals; LP, petal length (mm); WP, petal width (mm); LS, style length (mm); NS, number of stamens.

## Statistical analysis

The results were analyzed using the SPSS 22.0 software program for Windows (SPSS Science, Chicago, IL, USA). The differences between cultivars ($P < 0.05$) found after analyzing the different parameters studied were evaluated by analysis of variance (ANOVA). The mean values measured for each parameter were used to perform: (a) a principal component analysis (PCA) and (b) a clustering of cultivars into similarity groups using Ward's method for agglomeration and the squared Euclidean distance as a measurement of dissimilarity.

## RESULTS AND DISCUSSION

The data showed that the characters studied were highly variable, not only among the different varietals, but also among the different varieties that comprised these groups
(Tables 2– 5). The morphometric characters that had the greatest variability were in general those related to the fruit, arils and seeds. These characteristics will now be presented. The average fruit weight (FW) oscillated between 325 g (varietal group ME) and 414 g (varietal group CRO).

These data shows that the average weight of Spanish pomegranates is less than the Turkish (*Caliskan & Bayazit*, *2013*) or Moroccan (*Martínez et al.*, *2012*) varieties. The average weight of the arils varied between 0.39 and 0.55 g, and the production between 56 and 62%, which is similar to values from Iranian (*Tehranifar et al.*, *2010*) or Turkish (*Caliskan & Bayazit*, *2013*) pomegranates, but below the Moroccan Sefri and Ounk Hman varieties (*Martínez et al.*, *2012*). The total soluble solids (TSS) varied between 12.6% (CRO groups) and 15.3% (MA group), which shows that the Spanish varieties are in general less sweet than other varieties such as the Turkish 'Eksi' (18.5%; *Caliskan & Bayazit*, *2013*). However, the Spanish varieties are less acidic than those found in other countries, as shown by acidity values that oscillated between 0.21 and 0.48%.

Other varieties such as "Jabal" from Oman (*Al-Said, Opara & Al-Yahyai*, *2009*), Iranian varieties (*Tehranifar et al.*, *2010*) or the Turkish 'Lifani 2' variety (*Caliskan & Bayazit*, *2013*) have greater acidity. The maturity index (MI) varied between 37.6 (varietal group PTO) and 72.2 (MO group), while in the Moroccan varieties these values usually oscillated between 37.4 and 77.6 (*Martínez et al.*, *2012*) and this range tends to be wider among the Turkish varieties, which oscillate between 3.4 and 65.4 (*Caliskan & Bayazit*, *2013*).

The leaf surface area (LS) results from our study oscillated between 7.35 cm$^2$ (varietal groups CRO and 8.60 cm$^2$ (MO group)); the flower diameter (FD) varied between 10.6 mm (varietal group CRO) and 17.0 mm (ADO group), while the number of stamens varied between 245.8 (ME group) and 348.5 (ADO group). When analyzing the leaf and flower data, a lower variability was seen among varietal groups as compared to the variables measured in fruits and arils, and therefore had less discriminating power.

The PCA results revealed the existence of a high amount of variability among different varietal groups and among the varieties within each group, according to different morphometric and chemical characteristics that were measured in this research work.

Therefore, the findings from the pomegranate genotype grouping after the PCA were mainly based on the first three PCs, which accounted for 53.75% of the variability observed, with 27.77% (eigenvalue, 9.99), 17.49% (eigenvalue, 6.30), and 8.49% (eigenvalue, 3.06) for PC1, PC2 and PC3, respectively (Fig. 2). We defined values above 0.20 as significant for important parameters (Table 6).

The most important variables integrated by PC1 were fruit weight (FW), length (FL1, FL2), diameter (FD1) and arils. The weights of PC1 for leaf and flowers characteristics were less important. PC1 mainly separated the cultivars by the shape and size of their fruits and arils, with the groups composed by the cultivars PTO, CRO, PTB1 and ADO being the ones that had the largest fruits and arils (Fig. 2A, 4th quadrant-4C), with the accession group ME being the one with the smallest sizes (Fig. 2A, 3rd quadrant-3C). Other more recent and similar studies have shown that component PC1, the weight and shape of the fruit, is one of the main variables that differentiate the pomegranate genotypes, as found in studies performed in Croatia (*Radunic et al.*, *2015*) and Turkey (*Caliskan & Bayazit*, *2013*).

Martinez-Nicolas et al. (2016), *PeerJ*, DOI 10.7717/peerj.2214

**Table 2** **Mean values of fruit characters of each varietal group.**

| Variable | ADO | | | CRO | | | MA | | | ME | | | MO | | | PTO | | |
|---|---|---|---|---|---|---|---|---|---|---|---|---|---|---|---|---|---|---|
| | Mean | Min | Max | Mean | Min | Max | Mean | Min | Max | Mean | Min | Max | Mean | Min | Max | Mean | Min | Max |
| FW | 409 | 207 | 747 | 414 | 219 | 824 | 358 | 172 | 584 | 325 | 125 | 613 | 366 | 147 | 609 | 407 | 191 | 753 |
| FD1 | 93 | 75 | 116 | 96 | 74 | 118 | 89 | 68 | 109 | 86 | 60 | 111 | 90 | 65 | 110 | 93 | 70 | 120 |
| FD2 | 19 | 13 | 27 | 20 | 15 | 25 | 21 | 15 | 30 | 21 | 9 | 32 | 21 | 12 | 32 | 19 | 11 | 30 |
| FL1 | 79 | 63 | 99 | 80 | 54 | 105 | 77 | 59 | 100 | 74 | 51 | 99 | 77 | 57 | 92 | 80 | 60 | 99 |
| FL2 | 97 | 72 | 115 | 98 | 75 | 120 | 92 | 68 | 110 | 91 | 67 | 112 | 93 | 66 | 109 | 97 | 74 | 117 |
| FL3 | 19 | 3 | 26 | 18 | 9 | 29 | 15 | 2 | 32 | 16 | 1 | 30 | 16 | 6 | 25 | 17 | 4 | 26 |
| Nc | 6 | 5 | 9 | 7 | 5 | 8 | 7 | 5 | 8 | 7 | 5 | 9 | 7 | 5 | 9 | 6 | 5 | 8 |
| PcMc | 143 | 75 | 214 | 155 | 83 | 272 | 154 | 80 | 265 | 142 | 60 | 289 | 150 | 68 | 247 | 150 | 73 | 272 |
| Ec | 3 | 2 | 5 | 3 | 1 | 6 | 4 | 2 | 6 | 4 | 1 | 8 | 3 | 1 | 6 | 3 | 1 | 6 |
| Rs | 64 | 39 | 76 | 62 | 48 | 72 | 56 | 40 | 74 | 56 | 37 | 72 | 58 | 40 | 71 | 62 | 43 | 73 |

**Notes.**

For explanation of character symbols, see 'Material and Methods.'

Martinez-Nicolas et al. (2016), *PeerJ*, DOI 10.7717/peerj.2214

**Table 3** **Mean values of aril, seed and juice characteristics of each varietal group.**

| Variable | ADO | | | CRO | | | MA | | | ME | | | MO | | | PTO | | |
|---|---|---|---|---|---|---|---|---|---|---|---|---|---|---|---|---|---|---|
| | Mean | Min | Max | Mean | Min | Max | Mean | Min | Max | Mean | Min | Max | Mean | Min | Max | Mean | Min | Max |
| SW | 0.55 | 0.16 | 0.84 | 0.63 | 0.38 | 0.85 | 0.40 | 0.18 | 0.71 | 0.39 | 0.15 | 0.62 | 0.41 | 0.18 | 0.68 | 0.55 | 0.13 | 0.81 |
| SL | 12.8 | 7.2 | 15.8 | 12.7 | 8.4 | 16.4 | 10.3 | 6.2 | 13.3 | 10.3 | 1.2 | 14.2 | 10.4 | 5.2 | 13.8 | 12.4 | 6.5 | 17.2 |
| Sw | 7.22 | 3.50 | 10.23 | 7.66 | 4.90 | 10.22 | 6.63 | 3.40 | 9.84 | 6.48 | 1.89 | 11.15 | 6.31 | 2.99 | 10.79 | 7.20 | 2.61 | 11.74 |
| *l* | 7.16 | 0.41 | 9.99 | 7.38 | 4.65 | 10.62 | 6.40 | 2.98 | 10.23 | 6.11 | 2.10 | 9.96 | 6.02 | 3.14 | 9.04 | 7.71 | 3.46 | 14.88 |
| *w* | 1.94 | 0.43 | 3.41 | 2.24 | 0.90 | 3.93 | 2.21 | 0.34 | 4.37 | 1.96 | 0.20 | 4.54 | 1.72 | 0.48 | 4.17 | 2.12 | 0.13 | 4.44 |
| wpw | 0.04 | 0.02 | 0.08 | 0.05 | 0.01 | 0.09 | 0.04 | 0.01 | 0.08 | 0.04 | 0.01 | 0.10 | 0.04 | 0.02 | 0.09 | 0.04 | 0.02 | 0.08 |
| wpi | 7.75 | 3.30 | 18.20 | 7.98 | 1.37 | 15.82 | 10.23 | 3.63 | 21.01 | 10.14 | 2.18 | 22.05 | 10.09 | 4.31 | 21.45 | 7.75 | 3.30 | 18.20 |
| JV | 51.8 | 42.0 | 61.0 | 59.5 | 48.0 | 64.0 | 52.7 | 31.0 | 64.0 | 51.7 | 34.0 | 65.0 | 49.5 | 35.0 | 60.0 | 56.2 | 47.0 | 65.0 |
| pH | 3.78 | 3.11 | 4.10 | 3.93 | 3.82 | 4.01 | 3.98 | 3.62 | 4.23 | 4.05 | 2.62 | 5.94 | 4.03 | 3.94 | 4.11 | 3.72 | 2.89 | 4.10 |
| TSS | 13.7 | 12.0 | 15.2 | 12.6 | 10.9 | 13.2 | 15.3 | 13.8 | 17.0 | 14.6 | 12.3 | 19.8 | 14.7 | 12.4 | 16.9 | 14.0 | 12.0 | 16.3 |
| A | 0.42 | 0.26 | 0.90 | 0.31 | 0.28 | 0.35 | 0.23 | 0.20 | 0.31 | 0.24 | 0.17 | 0.92 | 0.21 | 0.16 | 0.28 | 0.48 | 0.26 | 1.46 |
| MI | 40.5 | 17.0 | 49.6 | 41.6 | 37.0 | 47.2 | 68.7 | 50.8 | 80.5 | 65.3 | 16.3 | 87.2 | 72.2 | 54.7 | 85.1 | 37.6 | 8.2 | 58.0 |

**Notes.**

For explanation of character symbols, see 'Material and Methods.'

**Table 4 Mean values of leaf and flowers characteristics of each varietal group.**

| Variable | ADO | | | CRO | | | MA | | | ME | | | MO | | | PTO | | |
|---|---|---|---|---|---|---|---|---|---|---|---|---|---|---|---|---|---|---|
| | Mean | Min | Max | Mean | Min | Max | Mean | Min | Max | Mean | Min | Max | Mean | Min | Max | Mean | Min | Max |
| LW | 19.9 | 11.3 | 30.8 | 19.1 | 12.9 | 26.9 | 22.4 | 13.6 | 31.0 | 21.1 | 9.8 | 34.0 | 22.4 | 10.5 | 33.6 | 20.0 | 11.4 | 33.3 |
| Ll | 53.2 | 28.6 | 89.0 | 51.3 | 26.1 | 74.7 | 54.7 | 33.9 | 89.0 | 51.5 | 24.1 | 88.9 | 54.1 | 22.9 | 85.9 | 50.6 | 22.4 | 87.4 |
| Lt | 58.6 | 34.2 | 95.8 | 56.6 | 31.6 | 82.6 | 60.8 | 38.5 | 95.3 | 57.1 | 27.7 | 97.3 | 60.0 | 28.6 | 95.4 | 55.6 | 27.6 | 97.1 |
| Lp | 5.38 | 2.42 | 9.40 | 5.31 | 1.83 | 10.01 | 6.11 | 1.86 | 10.73 | 5.58 | 1.65 | 10.63 | 5.90 | 2.05 | 10.67 | 5.03 | 1.65 | 10.61 |
| Ll/LW | 2.72 | 1.47 | 4.53 | 2.71 | 1.68 | 4.32 | 2.47 | 1.56 | 3.90 | 2.48 | 1.00 | 4.51 | 2.43 | 1.57 | 4.50 | 2.56 | 1.00 | 4.51 |
| LS | 7.47 | 1.37 | 16.77 | 7.35 | 2.58 | 17.43 | 8.35 | 0.50 | 16.75 | 7.75 | 0.30 | 18.07 | 8.60 | 3.03 | 17.34 | 7.75 | 1.32 | 17.40 |
| FD | 17.0 | 7.1 | 35.2 | 10.6 | 7.5 | 17.13 | 13.2 | 8.3 | 19.6 | 15.2 | 9.5 | 25.1 | 14.5 | 8.5 | 35.1 | 12.1 | 7.2 | 18.2 |
| FL | 27.0 | 13.9 | 39.7 | 31.7 | 24.4 | 46.1 | 31.1 | 21.4 | 44.0 | 26.5 | 10.3 | 41.7 | 29.9 | 12.8 | 44.4 | 33.5 | 15.8 | 48.5 |
| Np | 6.21 | 5.00 | 8.00 | 6.34 | 6.00 | 8.00 | 6.62 | 5.00 | 8.00 | 6.32 | 5.00 | 8.00 | 7.18 | 6.00 | 9.00 | 7.44 | 6.00 | 9.00 |
| Lp | 19.6 | 15.2 | 24.3 | 21.2 | 17.8 | 25.4 | 22.3 | 18.5 | 27.2 | 22.1 | 15.1 | 30.0 | 22.5 | 15.4 | 28.4 | 20.9 | 14.6 | 28.5 |
| Wp | 15.1 | 10.7 | 19.0 | 16.0 | 13.9 | 19.2 | 17.2 | 13.7 | 20.2 | 17.2 | 10.9 | 22.0 | 17.4 | 12.4 | 22.7 | 15.7 | 10.9 | 21.3 |
| Ns | 6.21 | 5.00 | 8.00 | 6.34 | 6.00 | 8.00 | 6.66 | 6.00 | 8.00 | 6.32 | 5.00 | 8.00 | 7.18 | 6.00 | 9.00 | 7.44 | 6.00 | 9.00 |
| LS | 12.2 | 2.7 | 28.8 | 12.4 | 2.9 | 28.2 | 19.2 | 5.2 | 30.8 | 12.5 | 1.4 | 39.2 | 16.4 | 2.8 | 30.8 | 13.4 | 3.0 | 28.7 |
| NS | 348.5 | 216 | 510 | 332.6 | 218 | 476 | 327.3 | 204 | 512 | 245.8 | 108 | 372 | 337.5 | 168 | 532 | 343.5 | 204 | 526 |

**Notes.**

For explanation of character symbols, see 'Material and Methods.'

**Table 5** Analysis of variance of each variable analyzed within each group of varieties studied.

| Fruit characteristics | | | | | | | | | |
|---|---|---|---|---|---|---|---|---|---|
| | FW | FD1 | FD2 | FL1 | FL2 | FL3 | Nc | PcMc | Ec | Rs |
| ADO | ** | * | ns | * | ns | * | ns | ns | ns | ** |
| CRO | ** | * | ns | ns | ns | ns | ns | ns | ns | * |
| MA | ns | ns | *** | ns | ns | ns | ns | ns | ** | ** |
| ME | *** | *** | *** | *** | *** | ** | *** | ** | *** | *** |
| MO | ns | ns | ns | ns | ns | ns | ns | ns | ns | ns |
| PTO | ** | *** | ** | ** | *** | ** | ns | * | ** | *** |

| Aril, seed and juice characteristics | | | | | | | | | | | |
|---|---|---|---|---|---|---|---|---|---|---|---|---|
| | SW | SL | Sw | *l* | *w* | wpw | wpi | JV | pH | TSS | A | MI |
| ADO | *** | *** | *** | * | *** | ** | *** | * | ns | ns | ns | * |
| CRO | ns | ** | ns | ns | ns | ns | ns | ns | ns | ns | ns | ** |
| MA | *** | *** | * | *** | *** | *** | *** | ns | ns | ns | ns | ns |
| ME | *** | *** | *** | *** | *** | *** | *** | ns | ns | ns | ns | ** |
| MO | ns | ** | ** | *** | * | *** | *** | ns | ns | ns | ns | ns |
| PTO | *** | *** | *** | *** | *** | *** | *** | ns | *** | ** | *** | *** |

| Leafs | | | | | | Flowers | | | | | | | |
|---|---|---|---|---|---|---|---|---|---|---|---|---|---|
| | LW | Ll | Lt | Lp | Ll/LW | LS | FD | FL | Np | Lp | Wp | Ns | LS | NS |
| ADO | *** | *** | ** | ns | ns | ns | *** | *** | ns | ns | *** | ns | ns | ns |
| CRO | ns | *** | *** | ns | ** | * | ns | ns | ** | ** | ns | ** | ns | ns |
| MA | ** | ** | ** | * | *** | ns | ns | *** | *** | *** | *** | *** | ns | *** |
| ME | *** | *** | *** | *** | *** | *** | *** | *** | *** | *** | *** | *** | *** | *** |
| MO | *** | *** | *** | *** | *** | * | *** | *** | *** | *** | *** | *** | *** | *** |
| PTO | *** | *** | *** | *** | *** | *** | *** | *** | *** | *** | *** | *** | *** | *** |

**Notes.**
*, **, *** and 'ns' indicate significant differences at $P < 0.05$, $P < 0.01$, $P < 0.001$ levels as well as non-significant, respectively. For explanation of character symbols, see 'Material and Methods.'

PC2 explains, overall, the rind weight plus the weight of carpellary membranes (and therefore aril yield), the woody portion index of the arils (seeds), the leaf area and the length of the fruit, flowers, petals and sepals, as well as juice acidity (pH and AT). Overall, this component differentiated the varieties by the acidity of their juice as well as their woody portion index. Figures 2A and 2C shows how varietals BO1 and BA1, which have a sour flavor, were grouped on the upper part of the first quadrant of the figure. Likewise, the varieties found in the first and second quadrant have a greater index of woody tissue.

PC3 integrated characters related with the shape and size of the flowers (FD, FL, NP, WP, Ns), leaf shape (LW, Ll/LW), skin thickness (Ec) and the maturity index (MI) (Table 6), although this component was less significant than PC1 and PC2. The other flower and leaf characteristics were not so important in the present study.

The cluster analysis produced a dendrogram with four main clusters (Fig. 3). The dissimilarity level (d) ranged from 1 to 25, revealing that there was a great degree of similarity/dissimilarity among varieties. The first cluster (I) included the ME group's

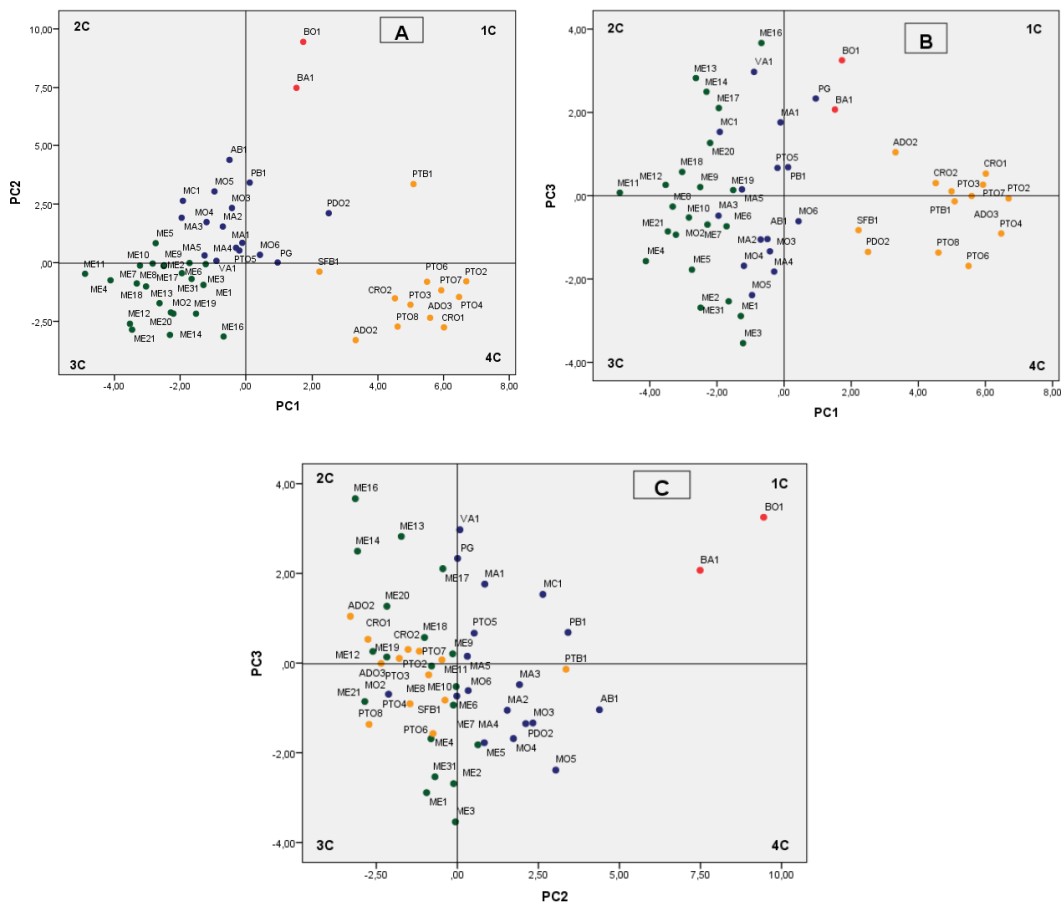

**Figure 2** **Plot of the principal components PC1–PC2 (A) PC1–PC3 (B) and PC2–PC3 (C) showing dispersion of Spanish pomegranates, based on morphological characteristics of the fruit and leaves, and the pH and acidity of the juice.** Each color indicates a group of varieties of similar characteristics.

cultivars (21 accessions), as well as the variety MO2, which was more similar to varieties from the ME group than to its own varietal group (MO). All of these fruits were medium sized (275.9–356.1 g), had a low-acidity juice, and high maturity indices in general (Table 3). As previously shown, the varieties from these groups were placed on the 3rd quadrant (3C) in the PC1 and PC2 principal component analysis graphic shown in Fig. 2A.

The second cluster (II) grouped cultivars BA1 and BO1, which were characterized by having medium-large fruit, high juice acidity and woody portion index. The dendrogram showed that these varieties were very similar, even though they came from different locations. These results can be found on the upper right part of the first quadrant (1C) (Figs. 2A and 2C).

The third cluster (III) was the most-heterogeneous group, as it was composed by 16 varieties from various locations, with fruits that were medium-large in size (331.5–436.5 g), and sweet juice (Tables 2 and 3). The dendrogram shows that there was a high degree of similarity between these 16 varieties, but at the same time, among these varieties, this similarity was greater between those that came from the same location or geographical area. These were mostly located in the second quadrant (2C) on Fig. 2A.

**Table 6** Eigenvalues, proportion of variation and eigenvectors associated with three axes of the PCA in pomegranate germplasm.

| Principal components (axes) | 1 | 2 | 3 |
|---|---|---|---|
| Cumulated proportion of variation | 27.77 | 45.26 | 53.75 |
| Characters | | Eigenvectors | |
| FW | **0.28** | 0.06 | −0.01 |
| FD1 | **0.28** | 0.07 | −0.01 |
| FD2 | −0.15 | 0.07 | −0.19 |
| FL1 | **0.26** | 0.09 | −0.03 |
| FL2 | **0.27** | 0.09 | 0.01 |
| FL3 | 0.11 | 0.03 | 0.11 |
| Nc | −0.12 | 0.02 | 0.15 |
| PcMc | 0.11 | **0.26** | −0.18 |
| Ec | −0.16 | 0.08 | **−0.24** |
| Rs | **0.22** | −0.17 | 0.17 |
| SW | **0.25** | −0.15 | 0.07 |
| SL | **0.27** | −0.15 | 0.05 |
| Sw | 0.19 | −0.17 | 0.09 |
| *l* | **0.25** | −0.02 | 0.05 |
| *w* | 0.11 | 0.07 | 0.13 |
| wpw | 0.13 | 0.12 | 0.11 |
| wpi | −0.13 | **0.25** | 0.01 |
| LW | −0.14 | 0.17 | **0.31** |
| Ll | 0.02 | **0.28** | −0.16 |
| Lt | 0.01 | **0.28** | −0.15 |
| Lp | −0.10 | 0.18 | 0.03 |
| Ll/LW | 0.12 | 0.07 | **−0.38** |
| LS | −0.01 | **0.22** | 0.15 |
| FD | −0.19 | 0.11 | **0.20** |
| FL | 0.17 | **0.21** | **0.23** |
| NP | 0.16 | 0.17 | **0.20** |
| LP | −0.10 | **0.24** | 0.12 |
| WP | −0.11 | 0.18 | **0.25** |
| Ns | 0.16 | 0.18 | **0.20** |
| LS | −0.01 | **0.29** | 0.14 |
| NS | 0.16 | 0.11 | −0.06 |
| JV | 0.12 | −0.01 | 0.15 |
| pH | −0.10 | **−0.22** | 0.22 |
| TSS | −0.10 | 0.08 | 0.14 |
| AT | 0.08 | **0.25** | −0.20 |
| MI | −0.19 | −0.12 | **0.23** |

**Notes.**
For explanation of character symbols, see 'Material and Methods.'

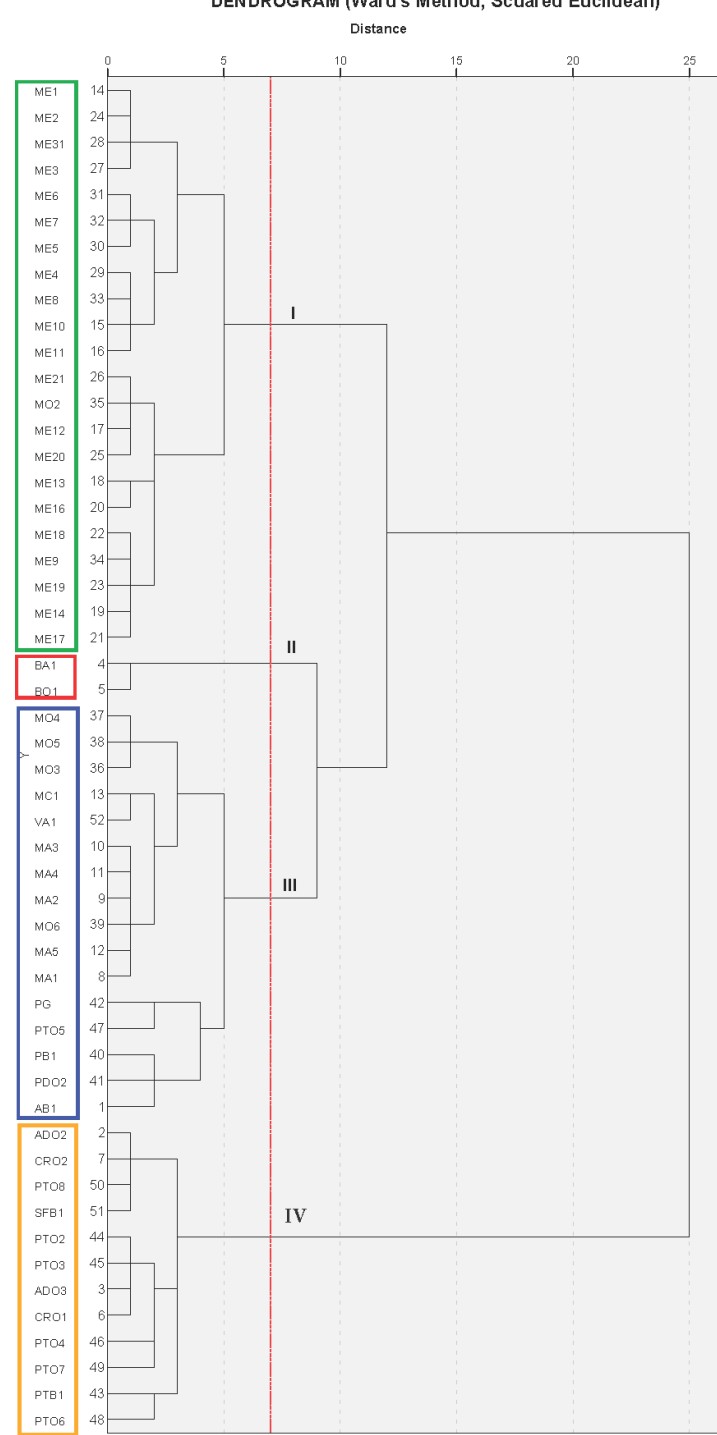

**Figure 3** **Cluster analysis grouping of 53 Spanish pomegranate cultivars (see Table 1 for cultivars names abbreviations).** Each color indicates a group of varieties of similar characteristics.
The last group (IV) was composed by 12 varieties, all of them from the same geographical area. As a whole, the varieties in this cluster were more similar to themselves than the varieties from clusters I and III (Fig. 3). The cluster IV varieties were characterized by their heavier fruit (358.8–464.2 g/fruit), and their large seeds (0.4–0.7 g/seed; Table 2). Most of the varieties from this group were placed in the fourth quadrant (Fig. 2A) and on the first and fourth quadrant of Fig. 2B, in the principal component analysis results.

In summary, this principal component and cluster analysis revealed two important results. First, in this pomegranate germplasm collection from Southeastern Spanish, there was a considerable variability among ascensions that may be mainly due to recombination (resulting from outcrossing) combined with sexual and vegetative propagation that occurred over a long period of time, as well as uncontrolled spread of plant material, as pomegranate is partially cross-pollinated (*Mars*, *1996*; *Jalikop & Sampath*, *1990*; *Martínez et al.*, *2009*). Second, within the group of cultivars 'ME,' 'MO,' 'MA,' 'PTO' or 'ADO,' a high degree of heterogeneity was observed.

It is therefore possible to think that all these groups could be "variety-population" (*Boulouha, Loussert & Saadi*, *1992*; *Tous, Battle & Romero*, *1995*; *Mars & Marrakchi*, *1999*), defined as plant material that although genetically different, have a certain degree of phenotypic resemblance. It is also interesting to point out that within the varieties analyzed, the four groups obtained in the cluster analysis (Fig. 3) coincided almost completely with their geographical origin. Therefore, in those cases where exchanges of plant material had not been the case, the geographical origin could be a determining criteria for the grouping of the varieties, except for MO2, BA1, BO1, PG and PTO5 (Fig. 3). This is in agreement with results reported for other fruit species (*Barbagallo, Lorenzo & Crescimanno*, *1997*), but also contradicts the grouping criteria obtained by *Mars & Marrakchi* (*1999*), where the geographical origin was not a determining factor for explaining the phenotypic variability of the pomegranate diversity in Tunisia. These authors have suggested that the geographical origin was not determinant because over time, there had been an exchange of plant material between the different growing areas. *Zhao et al.* (*2013*), in a study performed on 46 pomegranate cultivars, also indicated that cultivars were not clustered according to their morphological traits, agronomic traits, or geographic origin. Some authors have pointed to several causes for these inconsistencies, including: (1) the reproducibility of gene mutations caused the same mutation to emerge repeatedly in the distantly-related individuals from different areas (*Zhu*, *2002*); (2) the amplified polymorphic loci were not parts of the genes responsible for these morphological or the agronomic traits (*Jbir et al.*, *2008*; *Ebrahimi, Sayed-Tabatabaei & Sharifnabi*, *2010*) and (3) the quantitative traits were significantly influenced by the environment (*Zhu*, *2002*).

In this germplasm bank, no identical accessions were found within a single group, as shown in the ANOVA results on Table 5, as significant differences were found between the accessions belonging to a single group in most of the parameters analyzed, except for the juice characteristics, where differences in pH, TSS, TA and MI were observed only in the PTO group among its seven accessions. Among all the groups analyzed, ME was stood out, as there were significant differences in all the physical parameters measured in the fruits among its accessions. This evidences the great genetic diversity that exists even within a

single group, which could be explained not only by its different geographical origin, but also by the fact that it is native material that has been developing for many years, and has not suffered recombination with native material from other geographical areas. The data from this experiment also further confirmed the results from a previous study performed by *Melgarejo et al.* (*2009*) which evaluated the genetic diversity of pomegranate cultivars by using Restriction Fragment Length Polymorphisms (RFLP) and Polymerase Chain Reaction (PCR) techniques. Ten pomegranate accessions from the varietal groups Mollar de Elche, Mollar de Albatera, Mollar de Orihuela, Valencianas and Bordes were evaluated, resulting in different genetic profiles for the different groups as well as for the accessions within a single group. Other studies on pomegranate have shown the large genetic variability of this crop by using molecular techniques such as simple sequence repeats markers (SSR, *Ferrara et al.*, *2014*; *Pirseyedi et al.*, *2010*; *Hasnaoui et al.*, *2012*), random amplified polymorphic DNA (RAPD, *Hasnaoui et al.*, *2010*; *Narzary et al.*, *2009*), microsatellite markers (*Singh et al.*, *2015*), or single-nucleotide polymorphism (SNP) markers *Ophir et al.* (*2014*).

## CONCLUSIONS

As a result of the present study, we can conclude that among all the parameters analyzed, those related to fruit and seed size and the juice's acidity and pH were what had the highest power of discrimination, and, are therefore the most useful for genetic characterization studies of pomegranate germplasm banks. This is opposed to leaf and flower characteristics, which had a low power of discrimination. This germplasm bank, more specifically, was characterized by its considerable phenotypic (and presumably genetic) diversity among pomegranate accessions, with a greater phenotypic proximity existing among the varieties from the same geographical area, suggesting that over time, there has not been an exchange of plant material among the different growing areas. Also, within the same varietal group, a great variability was found, as no identical accessions were found. In general, knowledge on the extent of the genetic diversity found in the collection is essential for germplasm management. In this study, these data may help in the developing of strategies for pomegranate germplasm management and may allow for a more efficient use of this germplasm in future breeding programs for this species.

### Funding
The authors were supported by the Project on Genetic Resources, Preservation of Endangered Species: pomegranate and quince, Ref. RFP2012-00009-00-00, funded by INIA-MINECO and FEDER, which allowed for the maintenance of the pomegranate tree and quince collection used in this study. The funders had no role in study design, data collection and analysis, decision to publish, or preparation of the manuscript.

### Grant Disclosures

The following grant information was disclosed by the authors:

Genetic Resources, Preservation of Endangered Species: pomegranate and quince: RFP2012-00009-00-00.

INIA-MINECO.

FEDER.

### Competing Interests

The authors declare there are no competing interests.

### Author Contributions

- Juan J. Martinez-Nicolas conceived and designed the experiments, performed the experiments, analyzed the data, contributed reagents/materials/analysis tools, wrote the paper, prepared figures and/or tables, reviewed drafts of the paper.
- Pablo Melgarejo, Pilar Legua and Francisca Hernández performed the experiments, contributed reagents/materials/analysis tools, reviewed drafts of the paper.
- Francisco García performed the experiments, wrote the paper, reviewed drafts of the paper.

### Data Availability

The raw data has been supplied as Data S1, and the media data is contained in the tables in the article.

### Supplemental Information

Supplemental information for this article can be found online at http://dx.doi.org/10.7717/peerj.2214#supplemental-information.

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
