# Peer review of "Genetic diversity of pomegranate germplasm collection from Spain determined by fruit, seed, leaf and flower characteristics"

_PeerJ, doi:10.7717/peerj.2214_

## Round 0.1 · original submission · Major Revisions

All reviewers think that the study is of value, however the manuscript has to be improved according to the suggestions of all three reviewers. Please, note the annotated manuscript of reviewer 1.

In total 6 tables are presented, however only 5 tables are cited in the text. The ANOVA analysis is cited as table 5 in line 246, but it is presented as table 6.

The current tables 3,4, and 5 containing data should be given as supplements. Figure 2 can be deleted or presented as supplement.

·

Basic reporting

The manuscript is important for genetic diversity of morphological characteristics of pomgranate germplasm. Therefore, the results can be used for pomegranate genetic evaluations compared with morphological parameters. The manuscript standarts are acceptable for the journal. Some suggestions were showed on the manuscript.

Experimental design

The manuscript experimental design are acceptable.

Validity of the findings

No comments

Additional comments

The manuscript is critical for pomegranate germplasm characterization. Therefore, you can explaned to your pomegranate genotypes with morphological parameters within results section.

·

Basic reporting

This is a good study focused on morphological features in a small set of pomegranate germplasm and identified diverse and similar accessions. This work will be useful for future breeding work. Authors tried to infer the origin of pomegranate. This is just not possible. The experimentation and analysis of geographical origin is different. That part is out of focus. Please delete it. You can build up origin in introduction. In that case you need to use recently published papers in Molecular Genetics and Genomics and the other one in PLOS ONE. I recommend major revision of this manuscript.

Ophir R, Sherman A, Rubinstein M, Eshed R, Sharabi Schwager M, Harel-Beja R, Bar-Ya'akov I, Holland D (2014) Single-Nucleotide Polymorphism Markers from De-Novo Assembly of the Pomegranate Transcriptome Reveal Germplasm Genetic Diversity. PloS one 9 (2):e88998. doi:10.1371/journal.pone.0088998

Singh NV, Abburi VL, Ramajayam D, Kumar R, Chandra R, Sharma KK, Sharma J, Babu KD, Pal RK, Mundewadikar DM, Saminathan T, Cantrell R, Nimmakayala P, Reddy UK (2015) Genetic diversity and association mapping of bacterial blight and other horticulturally important traits with microsatellite markers in pomegranate from India. Mol Genet Genomics 290 (4):1393-1402

Experimental design

Analysis of morphological traits and resolving phenotypic diversity is well planned and supported. Analysis pertaining to geographical origin should be removed as it need different set of experiment.

Validity of the findings

As stated earlier

Reviewer 3 ·

Basic reporting

#Clear, unambiguous, professional English language used throughout.

The manuscript is a standard work of profiling a germplasm collection. Nonetheless this study is important to understand which of the pomegranate characters can distinguish among the EPSO varieties in Southeastern Spain. This objective was written well and clear. In spite of that, there were some inconsistent with the PeerJ standards and style.

#Intro & background to show context. Literature well referenced & relevant.

The introduction is fair. However the literature is not well covered and some of the statements are lack of reference.
Line 68 and 71: The origin of pomegranate and its ability to be spread in arid areas should be followed by the proper references.
The main subject of this manuscript is the characterization a germplasm collection. Through all the manuscript only four studies are mentioned: two of phenotypic characteristic and two that are based on genetic markers. I can easily count (by PubMed and google scholar search) at least four recent studies from India, China, and Israel using RFLPs, SSRs, and SNPs that describes the genetic diversity of pomegranate collections. Those should be referred at least in the discussion.

#Structure conforms to PeerJ standard, discipline norm, or improved for clarity.

Abstract – background section in the abstract should contain the motivation of the study rather than the objective only. That can be done by shorten the ‘conclusion’ section (or discussion as it should be by PeerJ guidelines) which is too long. In the methods section the sentence “…phenotypic and genetic diversity in the local pomegranate germplasm” is misleading. Only phenotypic diversity was detected. The genetic diversity is a deduction of the phenotypic due to the experimental design in this study. Please rephrase.

#Figures are relevant, high quality, well labelled & described.

Figures are relevant however no legend was written although it is required. Moreover, in the text the authors address the reader to table 1, while they describe the PCA and cluster graphs, to memorize the cultivar locations. This is quite confusing way to conclude. Alternatively, using color or symbol code in the graphs with appropriate legend would be a standard and concise way to interpret the result. I strongly recommend of doing so.

#Raw data supplied (See PeerJ policy).

OK

Experimental design

#Original primary research within Scope of the journal.

It is original study and follows objective scientific standards


#Research question well defined, relevant & meaningful. It is stated how research fills an identified knowledge gap.

The question is well defined although it should be more specific by rephrasing the “…the objective of this research was to determine the genetic variability that existed among all the different genotypes,…” to “…the objective of this research was to determine the genetic variability that existed among all the different genotypes of Southeastern Spanish Pomegranate …”.

#Rigorous investigation performed to a high technical & ethical standard.

OK

#Methods described with sufficient detail & information to replicate.

The authors chose to analyze the phenotypic data using three methods: PCA, agglomerative clustering, and ANOVA. The way they interpreted the clustering is somewhat redundant to the PCA interpretation since they refer to one level only. This fact is expressed in the manuscript as every cluster in the hierarchical clustering is reflected in the PCA, for example see “As previously shown, the varieties from these groups were placed on the 3rd quadrant in the PC1 and PC2 principal component analysis graphic shown in figure 3. What about the other levels of the hierarchical clustering? What do they tell us?
Figure 2 can be removed. It adds no information to our knowledge.

First paragraph of the results and discussion is very important, i.e., the factors that correlate to the principal component. The authors refer to table 2 to observe the correlations. I see no correlation coefficient in the table and I don’t understand how to extract them.
I see no reason when the authors refer to three PCs why they do not to show the effect of those components in the PCA graph, i.e., plotting PC1 against PC3 and PC3 against PC2. That will probably give us more information and the fact whether the PC3 is significant to some of the characters “PC3 integrated characters related with the shape and size…” and not for the most of them would be clearer.

Validity of the findings

The findings are valid. However it is not clear which of the data were used to create the PCA and the dendrogram, whether it was table3, 4, or 5 or a merged table from all of them.

Additional comments

Minor revision:

Through all the manuscript dendogram should be corrected to dendrogram.

Line 121: “a space” please rephrase
Line 163: “The length were” please rephrase
Line 184: “…based on the first three PCs, which accounted for 53.75% of the variability observed, i.e. for 27.77%, 17.49% and 8.49% respectively”. Respective to what? No PC list was given. I guess it is in respect to the PC order from 1 to 3 but is vague. Please rephrase.
Line 265: “This evidences” please rephrase
Line 263: 7 should be seven.
Line 232: “… three issues.” There is First, Second, but no Third. I guess “It is also interesting..” is the third?
Inconsistency in the PCA plot’s quandrant. Once it is described as “4th quadrant, bottom right” and once as “second quadrant”. It is the best to put numbers inside the plot’s quandrants and refer to those numbers.

---

## Round 0.2 · Minor Revisions

The revised manuscript has improved considerably. There is still the request for some minor changes, which are mostly of formal nature by reviewer 3.

Please, note that Table 2 has been cited only after tables 3-6 in line 224. Please correct numbering of tables.

Reviewer 3 ·

Basic reporting

Figure 2 legend. Please use the word plot instead of biplot. Biplot is a plot that illustrates (in your case) both the scattering of the pomegranate variants and the pomegranate characteristics as vectors on the same plot (see https://en.wikipedia.org/wiki/Biplot). This would be very informative if you had plotted such a graph but you have not.
Figure 2 and 3. No explanation of the color code neither in the legend nor anywhere else. It seems that you have used the color code to connect the clusters in the dendrogram and the PCA plot rather than illustrating locations in table1. That should be specified, at least, in the figure legends.

Experimental design

No Comments. corrected as suggested.

Validity of the findings

No Comments. corrected as suggested.

Additional comments

Line 223: values should be corrected to absolute values
Lines 304: remove the word “to”

---

## Round 0.3 · accepted · Accept

The authors have completed all requested changes.